# Exploring Subjective Happiness, Life Satisfaction, and Sustainable Luxury Consumption in China and Japan Amidst the COVID-19 Pandemic

Kyung-Tae Lee [1,*] and Hiroyasu Furukawa [2]

1   Department of Marketing and International Trade, Faculty of Commerce, Chuo University, Tokyo 192-0393, Japan
2   School of Business Administration, Meiji University, Tokyo 101-8301, Japan
*   Correspondence: klee097@g.chuo-u.ac.jp

**Abstract:** The COVID-19 pandemic has significantly impacted societies worldwide, leading to challenges in ensuring well-being and sustainability. In this context, it is essential to better understand consumer consciousness of pro-environmental products from the perspective of well-being. Therefore, our cross-national investigation aimed to explore the relationship between subjective well-being (SWB) and sustainable luxury consumption during the pandemic. We analyzed data from 800 respondents in China and Japan during the pandemic. Our findings, obtained through partial least squares structural equation modeling, reveal novel insights. First, SWB positively influences pro-environmental behavioral intentions, even in luxury fashion consumption. This highlights the potential for individuals to make sustainable choices, while indulging in luxury products. Additionally, we observed variations in the impact of subjective happiness and life satisfaction on luxury consumption between China and Japan. Specifically, the influence of subjective happiness was more pronounced in China than in Japan, indicating cultural distinctions in the relationship between well-being and luxury consumption. Moreover, our study identifies consumer novelty seeking as a mediator linking SWB to sustainable luxury consumption. This discovery provides a further understanding of the relationships explored in our study. These findings deepen our understanding of the interplay between well-being and sustainable luxury consumption, thereby informing strategies to promote pro-environmental behaviors in the luxury fashion sector.

**Keywords:** subjective happiness; life satisfaction; subjective well-being; consumer novelty seeking; purchase intention of sustainable luxuries; COVID-19

## 1. Introduction

The prolonged COVID-19 pandemic has threatened people's quality of life and brought about changes in values and lifestyles across countries (Echegaray 2021; Lee 2022). Furthermore, the impacts on consumer well-being and behavior are mixed. While the pandemic has reduced the consciousness of sustainable consumption, which benefits well-being (Hüttel and Balderjahn 2022), it has encouraged consumers to spend money to achieve more intrinsic well-being rather than pursuing material goals (Moldes et al. 2022). Considering the recent shaky relationship between well-being and sustainable behaviors, this study aims to elucidate how multiple types of subjective well-being (SWB) are related to environmentally sustainable consumption, while also considering the mediating role of consumer novelty seeking (CNS). Specifically, shedding light on luxury consumption, which has been described as being at odds or in affinity with sustainability (Athwal et al. 2019; Kunz et al. 2020), we conducted a cross-national study on the relationship between SWB and the consumption of pro-environmental luxury goods. This relationship is expected to vary across countries. For example, emerging countries with rapidly growing economies are likely to have high environmental footprints. Consumers in these countries

may feel guilty about their environmental burden and may be more motivated to consume pro-environmental luxury goods than in other countries.

For consumers who engage in sustainable behaviors, SWB is complementary to ecologically responsible behaviors and its extent concerns sustainable consumption (Brown and Kasser 2005). This finding aligns with the view that people generally engage less in behaviors that benefit others and society unless they are primarily happy (Diener et al. 2018; Nguyen et al. 2022). Although some exceptions exist, such as Horike (2012), who found a weak correlation and non-significant causal relationship between SWB and sustainable behaviors, previous studies have shown mutually reinforcing relationships between the two constructs. Specifically, pro-environmental behavior bolsters SWB (Capstick et al. 2022; Kaida and Kaida 2015; Schmitt et al. 2018; Zawadzki et al. 2020); conversely, SWB encourages pro-environmental behavior (Kushlev et al. 2020; Wang and Kang 2019; Yakut 2021) and ecologically responsible conduct (Brown and Kasser 2005). We extend the current knowledge by empirically addressing the following issues, focusing on sustainable consumption driven by SWB.

First, we investigated whether the positive relationship between SWB and sustainable consumption persisted in the context of conspicuous luxury consumption during the pandemic, posing threats to both well-being and sustainability. While previous research suggests that SWB is positively related to sustainable consumption, it remains unclear whether SWB plays a significant role in consuming luxury products that emphasize status and prestige. The conspicuous nature of luxury products tends to appeal to materialistic consumers (Sun et al. 2014), who not only prioritize possessions and are at odds with SWB (Ohno et al. 2022), but also prefer eco-friendly luxury products to display their status and prestige socially (Furukawa and Lee 2020). Furthermore, luxury consumption is often perceived as incompatible with sustainability (Kunz et al. 2020). However, given that sustainable luxury products are socially desirable and foster positive brand associations (Kumagai 2020), those with higher levels of well-being may be more inclined to support sustainability by choosing pro-environmental products for luxury consumption. With this in mind, we investigated the relationship between SWB and the purchase intention of sustainable luxury products (PISL).

Second, we focused on subjective happiness and life satisfaction to assess SWB. Life satisfaction, which represents the cognitive aspect of SWB (Diener 1984; Diener et al. 2018), is one of the most widely adopted constructs for exploring the relationship between SWB and sustainable behavior. However, limited attention has been paid to subjective happiness, which partially overlaps with life satisfaction but views SWB from a different angle (Eldeleklioğlu 2015; Iani et al. 2014; Lyubomirsky and Lepper 1999). Subjective happiness refers to a person's subjective assessment of their overall happiness, which includes cognitive and affective aspects and can be compared with others or based on self-characterization as happy or unhappy individuals (Lyubomirsky and Lepper 1999). Thus, we attempted to provide new insights into the relationship between SWB and PISL by shedding light on both subjective happiness and life satisfaction.

Third, we explored the mediating role of novelty-seeking tendencies in the relationship between SWB and PISL. Although SWB is known to affect sustainable behavior, its mechanism remains unclear. This novelty-seeking tendency concerns the theory of optimal stimulation level (OSL) (Raju 1980), wherein people are not satisfied with too many or too few stimuli in their daily lives and act in pursuit of an optimal stimulus (Raju 1980). In particular, intrinsic motivations to seek OSL lead to curiosity-based behaviors such as novelty seeking and variety seeking (Furukawa and Lee 2020; Raju 1980). From the perspective of consumer behavior, Manning et al. (1995) proposed CNS, which suggests that consumers seek novel product information to obtain optimal stimuli that satisfy their curiosity. We adapted CNS to capture the general tendency of behavior in the search for novel information on sustainable products, predicting that such a CNS tendency would mediate the effects of SWB on PISL.

Finally, we conducted a cross-national study in China and Japan. Asia is the most significant contributor to global luxury consumption, with Asian countries and Asian consumers accounting for 34% and 56% of the global luxury market in 2019, respectively (Bain & Company 2022). China and Japan have a strong presence in Asian luxury markets. According to Bain & Company (2022), the global luxury goods market shrank in 2020 due to the COVID-19 pandemic, but has recently enjoyed a V-shaped recovery since 2021. Chinese and Japanese consumers accounted for 33% and 10% of the global luxury market, respectively. Therefore, consumers in these countries are more likely to purchase luxury goods and, in such circumstances, perceive the difference in preferences between ordinary and sustainable luxuries more readily. Furthermore, previous studies have suggested that Japanese consumers are less environmentally conscious and feel less guilty about their environmental impact, whereas Chinese consumers are the opposite (Greendex 2014; Haerpfer et al. 2022). As guilt pertains to overall well-being and serves as a motivator for purchasing pro-environmental luxury goods (Sun et al. 2022; Wang et al. 2021), it may provide clues for predicting dissimilarities in the relationship between SWB and PISL in these two countries.

As mentioned previously, the objective of this study is to clarify the relationship between subjective happiness, life satisfaction, and CNS with sustainable luxury consumption through a cross-national investigation between China and Japan. To achieve this purpose, we set the following questions as specific research topics and attempted to answer them empirically.

1.  Are the effects of SWB, consisting of subjective happiness and life satisfaction, on sustainable consumption replicated in the context of luxury consumption during the COVID-19 pandemic?
2.  How does CNS mediate the effects of subjective happiness and life satisfaction on PISL?
3.  How are these relationships different or similar in China and Japan?

Elucidating the above research questions will provide novel insights that are yet to be thoroughly examined, thus contributing to the existing research.

The rest of this paper is organized as follows: Section 2 provides a literature review and proposes the hypotheses and conceptual models. Section 3 describes the research design and data collection. Section 4 presents data analysis and results. Section 5 discusses the findings and implications and outlines the limitations and suggestions for future research. Finally, Section 6 presents the conclusions of this study.

## 2. Conceptual Background and Hypotheses

### 2.1. SWB

Well-being refers to "a state of flourishing that involves health, happiness, and prosperity" (Mick et al. 2012, p. 6). Psychologists consider two main perspectives (Asano et al. 2014; Deci and Ryan 2008; Lee 2019; Ryan and Deci 2001) that are complementary to realizing well-being (Asano et al. 2014): the eudaimonic and the hedonic approach. The former considers well-being as living well through personal growth and self-actualization (Deci and Ryan 2008); it is a process of developing one's potential and achieving self-actualization (Deci and Ryan 2008; Ryan and Deci 2001). On the other hand, the latter regards well-being as a state in which physical and mental pleasure is satisfied and pain is avoided (Deci and Ryan 2008; Ryan and Deci 2001). Research using the hedonic approach has chiefly captured the degree of well-being based on self-rated SWB (Ryan and Deci 2001), representing a subjective sense of well-being and fulfillment as assessed using self-criteria (Diener 1984; Diener et al. 1985). In this regard, SWB is often used synonymously with happiness in mainstream happiness research (Diener 1984; Deci and Ryan 2008). The concepts of SWB and happiness can vary significantly depending on culture and society. Different societies and cultures assign varying degrees of importance to different types of SWB and happiness, and the factors that contribute to SWB differ across cultures (Diener et al. 2018). However, it is essential to note that certain universal factors that contribute to SWB, such as good health

and social relationships, also exist (Diener et al. 2018; Ohno et al. 2023). Prior research has shown that SWB is negatively related to materialistic tendencies (Brown and Kasser 2005; Lee 2019; Ohno et al. 2022) and feelings of personal relative deprivation (Ohno et al. 2023). Specifically, the pursuit of happiness through materialistic consumerism tends to prioritize external goals and may run counter to SWB, giving rise to materialistic desires and anxiety about status and financial success (Dittmar et al. 2014; Furukawa and Lee 2023; Kasser 2018).

### 2.2. Subjective Happiness and Life Satisfaction

Subjective happiness is "a global, subjective assessment of whether one is a happy or an unhappy person" (Lyubomirsky and Lepper 1999, p. 139). Although subjective happiness and life satisfaction overlap and correlate, their perspectives are somewhat different (Eldeleklioğlu 2015; Iani et al. 2014; Lyubomirsky and Lepper 1999). Life satisfaction is a cognitive, judgmental aspect of SWB (Diener et al. 1985; Diener et al. 2018), referring to the extent to which a person positively evaluates their overall quality of life (Veenhoven 1996) or a global assessment of their quality of life according to their own success criteria (Shin and Johnson 1978). By contrast, subjective happiness refers to whether one considers oneself to be a happy person by comparison with others or by evaluating whether one has the characteristics that happy people possess (Lyubomirsky and Lepper 1999). For example, one person may be satisfied with their life because they have a job, house, family, etc., but not feel happy; meanwhile, another person may have a low quality of life and sometimes feel negative emotions, but think of themselves as happy (Iani et al. 2014; Lyubomirsky and Lepper 1999). Empirical studies have demonstrated that the two constructs are distinct despite positive correlation coefficients of 0.40 (Eldeleklioğlu 2015), 0.59 (Iani et al. 2014), or 0.64 (Purvis et al. 2011).

Previous studies have shown that happy people tend to consume sustainably, shedding light on several aspects of well-being, including life satisfaction (Wang and Kang 2019; Yakut 2021), intrinsic satisfaction (De Young 2000), single-item SWB (Brown and Kasser 2005), and psychological and social well-being (Nguyen et al. 2022). Happy individuals engage in sustainable consumption because they are more intrinsically oriented and mindful of viewing others positively (Brown and Kasser 2005), actively participate in activities that benefit both society and themselves (Diener et al. 2018; Nguyen et al. 2022), and are relatively unlikely to cause hedonic adaptation (Capstick et al. 2022). Conversely, unhappy people are more preoccupied with personal worries and concerns and less likely to participate in social issues (Nguyen et al. 2022).

Regarding the effects of subjective happiness and life satisfaction on PISL, we postulated their relationships based on the theories of warm-glow giving and helper's high, which explain the utility of altruistic behavior.

The warm-glow-giving theory divides altruism into pure and impure (Andreoni 1990). The former refers to helping others without requesting compensation. The latter indicates altruism, which is expected to increase satisfaction through helping others. The act of giving derived from altruism is called warm-glow giving, which involves impure altruism characterized by altruistic and egocentric motivations aimed at enhancing self-satisfaction (Andreoni 1990). Happy people may have acquired a warm glow, and such behavior toward others is rewarded as self-satisfaction. As sustainable behaviors can be considered altruistic, happy people are likely to engage in sustainable consumption to gain a warm glow.

Similarly, the theory of helper's high also describes the relationship between SWB and PISL. This theory states that selfless service for others produces positive emotions and mood improvements (Baraz and Alexander 2010; Dossey 2018). People with high subjective happiness and life satisfaction are likely to realize that selfless services make them happy. Thus, they are likely to continue to act for others to maintain and strengthen their happiness. Therefore, the higher subjective happiness and life satisfaction, the more likely they are to be optimistic about sustainable consumption.

**H1.** *(a) Subjective happiness and (b) life satisfaction positively relate to PISL.*

However, the effects of SWB on PISL are likely to differ in China and Japan. Rapid economic growth has led to environmental degradation, and consumers in various countries are beginning to feel guilty about environmentally burdensome behaviors. It is argued that feelings of guilt facilitate prosocial behaviors (Abbate et al. 2022; Xu et al. 2012). Greendex (2014) examined and ranked the degree to which consumers in different countries felt guilty about environmentally damaging activities. Among the top 20 global luxury markets, China had the highest number of consumers who felt guilty about the environmental burden, whereas Japan had the fewest. Indeed, a global survey conducted by the World Values Survey (Haerpfer et al. 2022) also found a difference in environmental awareness between the two countries, with 68% of respondents in China (n = 3036) placing more value on protecting the environment than on economic growth, compared to 34% in Japan (n = 1353).

Considering this perspective of guilt concerning environmental awareness, the extent to which SWB encourages pro-environmental behaviors will be more pronounced in those who perceive themselves as happy compared with others in the social context; that is, those with high subjective happiness who judge their degree of well-being in comparison to others. This is because people with high subjective happiness are more likely to attempt to reduce their feelings of environmental guilt to maintain their current state of happiness, consequently engaging in altruistic pro-environmental behaviors. We expect similar behavior for sustainable luxury goods. That is, by choosing environmentally conscious luxury, one with high subjective happiness would not only obtain a high level of warm glow and helper's high, but also mitigate the feeling of environmental guilt. As a result, the impact of subjective happiness on PISL may be more prominent in China, where people perceive greater environmental responsibility than in Japan.

By contrast, life satisfaction captures the cognitive aspects of SWB (Diener 1984; Diener et al. 1985). People with high life satisfaction are likely to cognitively attempt to experience satisfaction, such as a warm glow and helper's high, to enjoy their current quality of life. This is likely to be similar across countries and cultures. Accordingly, regardless of the country, consumers with high life satisfaction are likely to show pro-environmental purchasing intentions regarding luxury consumption.

Thus, we propose the following hypotheses:

**H2.** *The effect of subjective happiness on PISL is more prominent in China than in Japan.*

**H3.** *The effect of life satisfaction on PISL is not significantly different between China and Japan.*

*2.3. CNS*

According to the OSL theory, people live with various stimuli but are not satisfied with too many or too few stimuli; hence, they seek an optimal level of stimulation (Furukawa and Lee 2020; Raju 1980). Novelty seeking is an exploratory behavior that looks for novel information that meets the optimal level of stimulation (Hirschman 1980). Hirschman (1980) argues that novelty seeking consists of two components: inherent novelty seeking, which represents a person's desire to seek novel stimuli, and actualized novelty seeking, which represents the actual behavior of acquiring novel stimuli. Manning et al. (1995) elaborate on Hirschman's (1980) novelty seeking in the context of consumer behavior. They proposed the concept of CNS, which sheds light on the behavior of seeking novel product information. Recent research has found that an intrinsically motivated CNS facilitates variety seeking or exploratory acquisition of products (Furukawa and Lee 2020). Variety seeking refers to the behavioral tendency to switch between brands within a category to find diverse product and service experiences (Baumgartner and Steenkamp 1996; Kahn et al. 1986).

Focusing on the CNS tendency toward novel information about sustainable products, we speculate that such a tendency mediates the effects of subjective happiness and life

satisfaction on PISL. As mentioned previously, happy people are optimistic about sustainable consumption (Brown and Kasser 2005; De Young 2000; Wang and Kang 2019; Yakut 2021). Therefore, they are likely to search extensively for novel information on pro-environmental consumption. Consequently, this general tendency of CNS triggers interest in pro-environmental luxury, thereby mediating the effect of SWB. The argument of identity theory on self-continuity and self-sameness provides a plausible theoretical rationale for this prediction. Previous research has found that SWB and mental health are better for those who maintain their identity uniformly and continuously through various interpersonal interactions (Van Hoof and Raaijmakers 2002). Therefore, it seems reasonable for happy people to attempt to maintain a state of well-being by purchasing products and acting in a way that preserves the sameness and continuity of their ego identities. The same is true for sustainable consumption. People who feel happy are likely to maintain self-consistency by becoming optimistic about their eco-conscious consumption. This tendency is also manifested in luxury consumption through the purchase of pro-environmental luxuries rather than ordinary luxuries. By behaving in such a way that they exhibit a uniform and continuous ego identity, they will attempt to maintain or improve their SWB. Thus, we expect subjective happiness and life satisfaction to increase CNS toward sustainable products, resulting in a positive PISL. Furthermore, the general tendency of CNS toward sustainable products is likely to remain the same by nationality or culture. Thus, we anticipate that the function of CNS in mediating the effects of subjective happiness and life satisfaction on PISL will be observed irrespective of the country.

**H4.** *CNS for sustainable products positively relates to PISL.*

**H5.** *CNS for sustainable products mediates the effects of (a) subjective happiness and (b) life satisfaction, irrespective of country.*

Figure 1 illustrates the conceptual model used in this study.

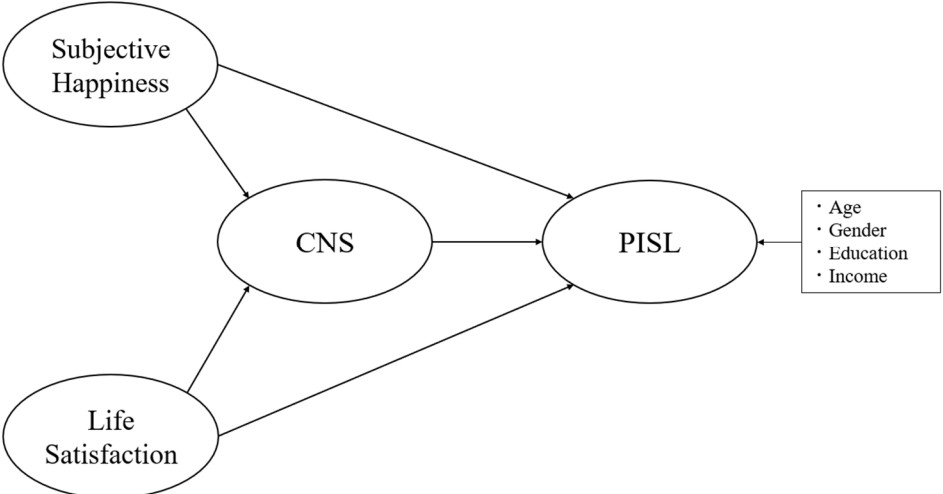

**Figure 1.** Conceptual model. Note: CNS = consumer novelty seeking for pro-environmental products; PISL = purchase intention of sustainable luxury.

## 3. Methods

### 3.1. Instruments

All construct items were measured using a seven-point Likert scale (1 = "strongly disagree", and 7 = "strongly agree"). We asked the respondents to indicate their degree of agreement with statements on subjective happiness, life satisfaction, CNS, and PISL. To assess subjective happiness and life satisfaction, we first employed Lyubomirsky and Lepper's (1999) Subjective Happiness Scale (SHS, n = 4) and Diener et al.'s (1985) Satisfaction with Life Scale (SWLS, n = 5). Second, we adopted Manning et al.'s (1995) CNS scale (n = 7) and modified it slightly to fit the context of novelty-seeking tendencies toward sustainable

products. Finally, PISL was measured using the three-item scale developed by Talukdar and Yu (2020).

The questionnaire items were translated into Chinese and Japanese as follows. Initially, the original English version was translated into Japanese using a back-translation method with the assistance of two bilingual speakers. However, we used previously validated Japanese versions for SHS (Shimai et al. 2004) and SWLS (Lee 2019). Subsequently, the completed Japanese version was translated into Chinese using the same-back translation method with the help of two other bilingual speakers. The translators confirmed that there were no issues of consistency or compatibility between the different versions.

*3.2. Data Collection*

Data were collected through online surveys conducted in July 2022 in China and Japan during the COVID-19 pandemic. To ensure the acquisition of high-quality data, online panel data obtained through a research agency were utilized. The research agency oversaw the sampling and data collection processes. First, in line with the policy for handling personal information, the following explanations are provided.

- Participation is voluntary.
- Anonymity is guaranteed.
- There are no right or wrong answers.
- Respondents give informed consent for using the data by answering the questionnaire.

These explanations reduce bias concerning social desirability and common methods (Podsakoff et al. 2003). Only respondents who agreed to the descriptions participated in the survey. We then asked the participants if they had experienced buying luxury fashion products. The fashion industry places a heavy burden on the environment, producing not only enormous amounts of waste, but also 20% of the world's wastewater and 10% of carbon emissions (UNEP 2019). In addition to their significant environmental impacts, fashion products often have conspicuous and self-expressive features, making them appropriate targets for this study. We explicitly instructed the research agency to screen the respondents with a period of purchase experience set to two years because it is critical to remember previous experiences clearly when answering questions. When respondents had multiple experiences, we asked them to answer only the experience they remembered most clearly. Finally, the participants answered the main questions and provided demographic information including age, sex, annual household income, and education level. The researchers did not use sensitive data to safeguard personal information.

We established a target sample size of 400 individuals per country, based on Israel's (1992) recommendation of a sample size of 400 with a 95% confidence level and a ±5% precision level when the population size is greater than 100,000 or is assumed to be infinite. Eligible participants were recruited until the sample size for each country reached 400, with no missing values in the data. Thus, 800 respondents completed the questionnaire: 400 from China and 400 from Japan. The respondents' ages ranged from 15 to 79 years (M = 40.56), with 16 to 73 years (M = 40.63) and 15 to 79 years (M = 40.45) in the Chinese and Japanese samples, respectively. The percentage of males among respondents was 45.8% (n = 183) in the Chinese sample, and 50.0% (n = 200) in the Japanese sample. The level of annual household income was as follows: for the Chinese sample, 21.5% had an income level below RMB 25,000; 33.3% responded between RMB 25,000 and RMB 59,999; 22% answered between RMB 60,000 and RMB 149,999; 18.5% had a high income level between RMB 150,000 and above; and 4.8% did not respond. In the Japanese sample, 19.3% had a lower income level of less than 3 million yen, 50.6% between 3 million yen and less than 8 million yen, 24.3% between 8 million yen and less than 1.5 million yen, and 1.3% had a high income level of 1.5 million yen and above. The remaining 4.8% did not respond. As of January 31, 2023, one dollar was approximately JPY 130 or RMB 6.8. Regarding education level, 71% (n = 284) of the Chinese respondents and 50.5% (n = 202) of the Japanese respondents had at least a college education.

### 3.3. Analytical Methodology

Partial least squares structural equation modeling (PLS-SEM) was conducted using SmartPLS 4 to evaluate the measurements and structural models. SmartPLS, provided by SmartPLS GmbH in Oststeinbek, Germany, is software for data analysis based on the PLS-SEM method. PLS-SEM is distinguished from covariance-based structural equation modeling (CB-SEM) in that it adopts a variance-based approach to structural equation modeling. While CB-SEM helps test how well a theoretical model fits the data, PLS-SEM is better suited for investigating the main factors explaining the target construct and for exploratory analysis to build a theoretical model (Hair et al. 2017). These characteristics were appropriate for this study, which investigated how well subjective happiness, life satisfaction, and CNS explain PISL in a cross-national context.

## 4. Analysis and Results

### 4.1. Measurement Model Assessment

In the measurement model, one item of subjective happiness and one item of life satisfaction were excluded because of inadequate factor loadings. Subsequently, we tested the measurement invariance of composite models (MICOM), which is necessary for undertaking a multigroup analysis in PLS-SEM (Hair et al. 2018; Henseler et al. 2016). A three-step approach was employed to analyze MICOM, consisting of configural invariance, compositional invariance, and equality of composite mean values and variances. The configural invariance was automatically established by running MICOM on SmartPLS 4 (Ringle et al. 2022). Compositional invariance was tested using permutation multigroup analysis (permutation MGA) with 5000 permutations. The permutation MGA returned a result in which the original correlation of life satisfaction (0.997) was smaller by 0.001 than that of the 5% quantile (0.998). Therefore, referring to the outer loadings, we deleted two items of life satisfaction and performed a permutation MGA again. The correlation scores were equal to or greater than the 5% quantile (Table 1), thereby establishing compositional invariance. Finally, we proceeded to the final step to examine the equality of composite mean values and variances. The permutation MGA showed that all the mean values and variances of CNS were significantly different between Japan and China. Accordingly, partial measurement invariance was established, allowing a comparison of the standardized coefficients across the two groups (Hair et al. 2018; Henseler et al. 2016).

**Table 1.** Results of MICOM.

| | Step 2 | | Step 3 (Mean) | | | Step 3 (Variance) | | |
|---|---|---|---|---|---|---|---|---|
| | Original Correlation | 5% Quantile | Original Difference | 2.5% | 97.5% | Original Difference | 2.5% | 97.5% |
| Subjective Happiness | 0.997 | 0.997 | 0.661 | −0.143 | 0.137 | −0.151 | −0.207 | 0.209 |
| Life Satisfaction | 1.000 | 0.999 | 0.718 | −0.136 | 0.139 | −0.205 | −0.204 | 0.208 |
| CNS | 1.000 | 1.000 | 0.947 | −0.137 | 0.138 | −0.454 | −0.219 | 0.21 |
| PISL | 1.000 | 1.000 | 0.932 | −0.141 | 0.142 | −0.129 | −0.217 | 0.205 |

The full collinearity assessment approach is recommended in PLS-SEM to assess common method bias (CMB) (Kock 2015). CMB is a bias resulting from the measurement method and not from theoretical models or constructs. A full collinearity test with a model in which a random variable was set as the endogenous variable showed that all variance inflation factor (VIF) values were below 3.3 (Kock 2015). Therefore, there is little concern regarding contamination by CMB.

As presented in Table 2, the measurement model indicated acceptable scores, meeting each recommended threshold for indicator reliability (IR > 0.50), composite reliability (CR > 0.70), and average variance extracted (AVE > 0.50) (Fornell and Larcker 1981; Garson 2016; Hair et al. 2017). As all AVE values were higher than 0.50, convergent validity was supported (Fornell and Larcker 1981; Garson 2016).

**Table 2.** Evaluation of the measurement model.

| Construct and Indicator | | China/Japan | | | |
|---|---|---|---|---|---|
| | | Factor Loadings | IR | CR | AVE |
| Subjective Happiness | In general, I consider myself: not a very happy person/a very happy person. | 0.819/0.849 | 0.671/0.721 | 0.712/0.857 | 0.635/0.728 |
| | Compared to most of my peers, I consider myself: less happy/more happy. | 0.798/0.832 | 0.637/0.692 | | |
| | Some people are generally very happy. They enjoy life regardless of what is going on, getting the most out of everything. To what extent does this characterization describe you? | 0.773/0.877 | 0.599/0.769 | | |
| Life Satisfaction | The conditions of my life are excellent. | 0.861/0.893 | 0.741/0.797 | 0.728/0.795 | 0.777/0.822 |
| | So far, I have gotten the important things I want in life. | 0.902/0.920 | 0.814/0.846 | | |
| CNS | I often seek out information about environmentally new products and brands. | 0.773/0.850 | 0.598/0.723 | 0.876/0.930 | 0.573/0.703 |
| | I like to go to places where I will be exposed to information about new environmental products and brands. | 0.770/0.845 | 0.593/0.714 | | |
| | I like magazines that introduce environmentally new brands. | 0.745/0.820 | 0.556/0.672 | | |
| | I seek out situations in which I will be exposed to new and different sources of environmental product information. | 0.749/0.840 | 0.561/0.706 | | |
| | I am continually seeking new environmental product experiences. | 0.765/0.873 | 0.585/0.762 | | |
| | When I go shopping, I find myself spending very little time checking out new environmental products and brands. | 0.754/0.797 | 0.569/0.635 | | |
| | I take advantage of the first available opportunity to find out about new and different environmental products. | 0.742/0.843 | 0.551/0.711 | | |
| PISL | Would you like to purchase sustainable luxury products (compared to generic luxury products) because the production is sustainable for the society and environment? | 0.833/0.862 | 0.694/0.743 | 0.774/0.803 | 0.688/0.717 |
| | Do you prefer to purchase sustainable luxury products rather than generic luxury products? | 0.816/0.834 | 0.666/0.696 | | |
| | Would you be willing to pay more to purchase sustainable luxury products because they are produced by a company with a focus on sustainability, eco-friendliness and socially responsible manufacturing (compared to generic luxury products)? | 0.840/0.844 | 0.706/0.712 | | |

We assessed discriminant validity using the Fornell–Larcker criterion, which proposes that the square root of AVE should be greater than each correlation coefficient between every two constructs to support discriminant validity (Fornell and Larcker 1981). As shown in Table 3, the Fornell–Larcker criterion was satisfied, thus supporting discriminant validity.

**Table 3.** Discriminant validity.

| | China | | | | Japan | | | |
|---|---|---|---|---|---|---|---|---|
| | Subjective Happiness | Life Satisfaction | CNS | PISL | Subjective Happiness | Life Satisfaction | CNS | PISL |
| Subjective Happiness | *0.797* | | | | *0.853* | | | |
| Life Satisfaction | 0.641 | *0.882* | | | 0.564 | *0.907* | | |
| CNS | 0.535 | 0.508 | *0.757* | | 0.276 | 0.445 | *0.839* | |
| PISL | 0.526 | 0.443 | 0.726 | *0.830* | 0.249 | 0.456 | 0.758 | *0.847* |

Note: **Bold italic** letters on the diagonal line denote the square root of the AVE, and the values under the diagonal line represent the correlation coefficients.

### 4.2. Structural Model Assessment

The structural model was evaluated using 5000 bootstrap samples. Furthermore, we conducted a multigroup analysis using PLS-MGA to compare China and Japan. Table 4 presents the results. The coefficients of determination ($R^2$), measuring the predictive power of the model, were distributed in the range of 0.199–0.600. All VIF values were lower than the cutoff value of 5, indicating no critical collinearity issues among the indicators (Garson 2016; Hair et al. 2017). The effect size ($f^2$) describes whether a specific exogenous construct has a substantive effect on endogenous constructs and requires a minimum value of 0.02 for a significant effect (Hair et al. 2017; Hair et al. 2019). For predictive relevance $Q^2$, a value larger than zero for a specific endogenous construct is suggested to have predictive accuracy for the structural model of the construct (Hair et al. 2017; Hair et al. 2019). All $Q^2$ values were greater than zero, indicating the predictive accuracy of the model.

**Table 4.** Structural model and PLS-MGA.

| | Path | | Path Coefficients | *t*-Value | $R^2$/Adj.$R^2$ | VIF | $f^2$ | $Q^2$ |
|---|---|---|---|---|---|---|---|---|
| | Subjective Happiness | $\longrightarrow$ CNS | 0.356 | 4.671 *** | 0.333/0.329 | 1.697 | 0.112 | 0.314 |
| | Life Satisfaction | $\longrightarrow$ CNS | 0.280 | 3.782 *** | | 1.697 | 0.069 | |
| China | Subjective Happiness | $\longrightarrow$ PISL | 0.183 | 3.051 ** | | 1.901 | 0.040 | |
| | Life Satisfaction | $\longrightarrow$ PISL | 0.002 | 0.027 | | 1.854 | 0.000 | |
| | CNS | $\longrightarrow$ PISL | 0.622 | 12.764 *** | | 1.557 | 0.566 | |
| | Age | $\longrightarrow$ PISL | −0.021 | 0.537 | 0.561/0.553 | 1.174 | 0.001 | 0.270 |
| | Gender | $\longrightarrow$ PISL | −0.050 | 0.702 | | 1.083 | 0.001 | |
| | Education | $\longrightarrow$ PISL | 0.082 | 2.047 * | | 1.137 | 0.014 | |
| | Income | $\longrightarrow$ PISL | −0.005 | 0.146 | | 1.153 | 0.000 | |
| | Subjective Happiness | $\longrightarrow$ CNS | 0.037 | 0.640 | 0.199/0.195 | 1.466 | 0.001 | 0.187 |
| | Life Satisfaction | $\longrightarrow$ CNS | 0.424 | 7.075 *** | | 1.466 | 0.153 | |
| Japan | Subjective Happiness | $\longrightarrow$ PISL | −0.038 | 0.930 | | 1.524 | 0.002 | |
| | Life Satisfaction | $\longrightarrow$ PISL | 0.158 | 3.046 ** | | 1.718 | 0.036 | |
| | CNS | $\longrightarrow$ PISL | 0.694 | 19.448 *** | | 1.258 | 0.959 | |
| | Age | $\longrightarrow$ PISL | −0.017 | 0.475 | 0.600/0.593 | 1.046 | 0.001 | 0.182 |
| | Gender | $\longrightarrow$ PISL | −0.138 | 2.134 * | | 1.027 | 0.012 | |
| | Education | $\longrightarrow$ PISL | 0.049 | 1.443 | | 1.103 | 0.006 | |
| | Income | $\longrightarrow$ PISL | 0.007 | 0.199 | | 1.077 | 0.000 | |
| | | | Differences of path coefficients (China–Japan) | | | | | |
| | Subjective Happiness | $\longrightarrow$ CNS | 0.318 ** | | | | | |
| | Life Satisfaction | $\longrightarrow$ CNS | −0.144 | | | | | |
| | Subjective Happiness | $\longrightarrow$ PISL | 0.221 ** | | | | | |
| | Life Satisfaction | $\longrightarrow$ PISL | −0.156 * | | | | | |
| PLS-MGA | CNS | $\longrightarrow$ PISL | −0.072 | | | | | |
| | Age | $\longrightarrow$ PISL | 0.004 | | | | | |
| | Gender | $\longrightarrow$ PISL | 0.088 | | | | | |
| | Education | $\longrightarrow$ PISL | 0.033 | | | | | |
| | Income | $\longrightarrow$ PISL | −0.012 | | | | | |

Note: * $p < 0.05$, ** $p < 0.01$, *** $p < 0.001$.

Moreover, multiple mediation analyses showed that, except for the indirect effect of subjective happiness on PISL via CNS in Japan, the roles of CNS in mediating the effects of subjective happiness and life satisfaction on PISL were significant across the two countries (Table 5).

**Table 5.** Mediation analysis.

| Path | China | | | Japan | | |
|---|---|---|---|---|---|---|
| | Path Coefficients | *t*-Statistics | *p* | Path Coefficients | *t*-Statistics | *p* |
| Subjective Happiness ⟶ CNS ⟶ PISL | 0.221 | 4.763 *** | 0.000 | 0.026 | 0.639 | 0.523 |
| Life Satisfaction ⟶ CNS ⟶ PISL | 0.174 | 3.567 *** | 0.000 | 0.294 | 6.532 *** | 0.000 |

Note: *** $p < 0.001$.

Table 4 shows that the effects of subjective happiness and life satisfaction on PISL are inconsistent across China and Japan. Although subjective happiness positively affected PISL in China, it did not significantly affect PISL in Japan. Although life satisfaction did not directly affect the PISL in China, it positively affected the PISL in Japan. Thus, both H1a and H1b are partially supported. The PLS-MGA results illustrate that the effect of subjective happiness on PISL is significantly more prominent in China than in Japan, in favor of H2. Contrary to our hypothesis, the effect of life satisfaction on PISL was significantly more pronounced in Japan than in China, leading to the rejection of H3. Moreover, CNS positively affected PISL in China and Japan, supporting H4. As is evident in Table 5, the mediation analysis indicated that, except for the indirect relationship between subjective happiness and PISL in Japan, CNS positively mediated the effects of subjective happiness and life satisfaction on PISL in the two countries, partially supporting H5.

## 5. Discussion and Implications

### 5.1. Discussion

The COVID-19 pandemic has impacted society and diversified individual values and lifestyles in various countries, resulting in a critical challenge in ensuring well-being and sustainability (Echegaray 2021; Hüttel and Balderjahn 2022; Lee 2022; Moldes et al. 2022). Therefore, it is imperative to better understand consumers' consciousness of pro-environmental products from the perspective of well-being. Hence, we conducted a cross-national comparative analysis of SWB and sustainable consumption, focusing on the luxury market that enjoys a V-shaped recovery. We focused on the context of fashion products because of the significant environmental impact of the fashion industry (UNEP 2019). Considering subjective happiness, life satisfaction, and CNS, the analysis derives novel findings by linking SWB, pro-environmental sustainability, and luxury fashion consumption. Although subjective happiness is one of the primary constructs representing SWB, its impact on sustainable consumption has yet to be fully validated. This study revealed that the role of subjective happiness in PISL varies across countries. Additionally, by introducing CNS as a mediator, we clarified the mechanism by which SWB leads to PISL. Thus, this study is the first to elucidate how multiple SWB constructs relate to CNS and the consumption of sustainable luxuries in a cross-national context.

Note that happy individuals' tendencies to buy sustainable products are replicated in the context of luxury consumption during the pandemic. This finding indicates that the logic of warm-glow giving and helper's high theories applies to the consumption of sustainable luxury goods. However, the results differed depending on the SWB type in the two countries. While life satisfaction significantly affected PISL directly or indirectly in both countries, subjective happiness impacted PISL in China but not in Japan. Therefore, the impact of SWB on PISL is not uniform across countries and varies depending on the SWB construct. Many studies have examined the relationship between a single SWB construct and sustainable behavior (Nguyen et al. 2022); however, cross-national or cross-cultural studies are rare. We address this paucity by clarifying that the relationship between SWB and PISL can be cross-nationally divergent according to the type of well-being.

The prominent effects of subjective happiness in China can be explained as follows. Subjective happiness involves the subjective evaluation of one's degree of happiness compared with others. For example, the items of subjective happiness include an assessment of SWB based on whether one considers oneself a happy person compared to others and whether one has the characteristics that happy people usually possess (Lyubomirsky and Lepper 1999). Furthermore, Chinese consumers tend to exhibit a heightened sense of guilt about their environmental impacts on a global scale, whereas Japanese consumers have relatively low guilt feelings associated with environmental issues (Greendex 2014; Haerpfer et al. 2022). Given these facts, a plausible explanation is that Chinese consumers with high subjective happiness who perceive happiness based on comparisons with others attempt to lessen the feeling of guilt more than others about environmental issues and maintain their current well-being by obtaining satisfaction, including warm-glow giving and helper's high, resulting in a significant relationship between subjective happiness and PISL. This interpretation aligns with the existing findings that guilt promotes prosocial behaviors (Abbate et al. 2022; Xu et al. 2012).

In Japan, subjective happiness had no significant relationship with PISL. This result is similar to that of Horike (2012), who surveyed Japanese respondents and found a weak relationship between SWB and sustainable behavior. As mentioned previously, Japanese consumers tend to feel less guilty about environmental impacts (Greendex 2014). Therefore, even when subjective happiness is high, consumers have less incentive to reduce their feelings of guilt about environmental impacts and achieve a perception of happiness, leading to a non-significant relationship between subjective happiness and PISL. However, we found that the effect of life satisfaction was more significant in Japan than in China. As mentioned, life satisfaction is a cognitive evaluation of the quality of life (Shin and Johnson 1978; Veenhoven 1996). Japanese consumers seem to act sustainably in luxury consumption only when they perceive their quality of life to be high to some extent. Thus, environmentally sustainable luxury consumption serves as an egoistic measure in Japan as it is used to preserve and fortify one's identity.

We further found that CNS for pro-environmental products mediates the effects of subjective happiness and life satisfaction on PISL, except for the impact of subjective happiness in Japan. Particularly in China, complete mediation by CNS indicates the importance of considering CNS in exploring the relationship between SWB and sustainable luxury. These findings suggest that the CNS tendency leads to a behavioral intention to value sustainability, even in conspicuous and materialistic luxury consumption. As mentioned in the proposed hypothesis, the self-continuity and self-sameness arguments in the identity theory allow us to understand this result well. People with happiness are apt to act in ways that benefit society and others (Brown and Kasser 2005; De Young 2000; Nguyen et al. 2022; Wang and Kang 2019). To consistently display a tendency to consider social desirability, they are likely to prefer pro-environmental products and search for novel information until they reach an optimal level of stimulation. Thus, the propensity to maintain self-continuity and self-sameness also appears in the context of luxury consumption and significantly affects PISL.

Therefore, this study clarifies how subjective happiness and life satisfaction lead to pro-environmental luxury consumption through CNS tendencies. Given that previous studies have mainly discussed the direct relationship between SWB and sustainable consumption, this study is the first to reveal the role of CNS in this relationship. The findings also suggest that the psychological tendency to maintain self-continuity/self-sameness and the variety-seeking tendency to seek optimal stimulus levels consistent with one's state of SWB are the appropriate theoretical underpinnings. Consequently, this study extends research on the relationship between SWB and pro-environmental behavior by elucidating the CNS-mediated mechanism in this connection based on identity and OSL theories. Furthermore, as China and Japan are representative examples of the largest luxury markets, the results of this study can serve as a valuable reference for conducting similar research in

other countries where luxury consumption is expected to expand in the future, along with economic growth, income improvement, and the convergence of the pandemic.

### 5.2. Implications

This study has the following implications. Theoretically, by encompassing insights from theories on warm-glow giving, helper's high, OSL, and identity, we propose and verify a theoretical model explaining the mechanism by which different types of SWB result in PISL through CNS in the cross-national context of China and Japan. It drew novel findings, such as the inconsistent effects of subjective happiness and the mediating effects of CNS on PISL. Simultaneously, by synthetically applying these theories, we confirm their extendibility to the notable phenomena of SWB and sustainable behavior.

Furthermore, we demonstrate that variations in the degree of consumer guilt regarding environmental burdens can contribute to the differential effects of subjective happiness between China and Japan. Thus, this study provides a unique perspective on disparities in pro-environmental consumption among consumers in different countries. As a result, this study provides an empirical account of the association of SWB with sustainable luxury consumption.

This study has the following practical implications: for sustainable luxury goods to succeed, luxury brands need to communicate novel information on how they can contribute to sustainability. It would also be helpful to persuade consumers that purchasing sustainable luxuries is consistent with their current happiness, maintaining continuity and sameness of identity. Particularly in China, those with higher subjective happiness, who generally reported higher well-being than others, were more likely to consume sustainable luxuries than ordinated luxuries because of their feeling of guilt about environmental burdens. Therefore, to promote sustainable luxury goods, it is beneficial to report that consuming sustainable luxury can help mitigate a negative effect on the environment, simultaneously showing the owners' status and social prestige.

However, in Japan, where subjective happiness is less effective, it would be helpful to emphasize personal and intrinsic values that can contribute to well-being in luxury consumption and bolster life satisfaction. When consumer identities align with a luxury brand's purpose and vision, consumers experience intrinsic satisfaction (Coelho et al. 2018). Therefore, advancing a brand's environmentally friendly purpose and vision likely positively affects intrinsic satisfaction, and thus stimulates pro-environmental consumption.

### 5.3. Limitations and Future Research

Despite these contributions, this study had several limitations. First, the data collection for this study was conducted via an online survey, meaning that only those with internet access could participate. Therefore, there are concerns that the representativeness and accuracy of the sample may not have been fully ensured. Future studies should employ diverse sampling methods to improve representativeness and devise ways to increase the accuracy of sample information. Second, we excluded three items of life satisfaction when conducting measurement invariance tests on the datasets collected in China and Japan. While it was inevitable to conduct this cross-national study thoroughly and precisely, life satisfaction still needs to be fully measured in future studies. Third, this study revealed that the function of subjective happiness in consuming pro-environmental luxury products diverges across countries. Future studies should quantitatively measure the degree of guilt and examine how guilt modulates the relationship between SWB and sustainable luxury consumption. Fourth, considering social, cultural, and personal antecedents, future studies should investigate how not only the cultural phenomenon of face-saving, which reportedly affects Chinese consumers' behaviors (Siu et al. 2016), but also social norms are associated with SWB and sustainable luxury consumption. This contributes to a better understanding of the cross-cultural relationship between consumer well-being and sustainable behaviors. Fifth, this study focused on luxury fashion products and did not consider other luxury or non-luxury categories. Given that consumer responses may vary by product type and that

there are markets to generate more waste, future studies should consider and compare other luxury and non-luxury goods. Finally, data were collected and analyzed during the pandemic. Therefore, it is crucial to investigate whether these findings can be replicated during non-pandemic periods. To achieve this, a longitudinal research approach should be considered, in which a similar study can be conducted after the COVID-19 pandemic has subsided, and the results can be compared to those obtained in the present study.

## 6. Conclusions

Due to the COVID-19 pandemic, it is vital to improve well-being and sustainability further. This study showed how subjective happiness and life satisfaction lead to buying intentions toward pro-environmental fashion luxury via CNS in China and Japan. The following conclusions can be drawn based on the results: SWB can lead to pro-environmental behavioral intentions, even when consuming luxury fashion products, and the theoretical rationales of warm-glow giving and helper's high plausibly explain this finding. However, the effects of subjective happiness and life satisfaction on luxury consumption vary between China and Japan. Specifically, the impact of SH is more pronounced in China than in Japan, which could be explained by the difference in consumer guilt regarding the burden placed on the environment. Additionally, CNS mediates the effect of SWB on sustainable luxury consumption, and the theories of self-continuity/self-sameness and variety seeking toward OSL provide a theoretical rationale to appropriately explain this result.

This study extends existing research by incorporating concepts and contexts that have not been explored in previous studies. Specifically, we proposed and tested a theoretical model to explain the mechanisms underlying these findings. By clarifying the different effects of subjective happiness and life satisfaction on PISL via CNS in China and Japan, this study contributes to a better understanding of the relationship between SWB and sustainable luxury consumption. In clarifying the relationship between SWB and people's sustainable behaviors in this way, this research has valuable implications not only for those involved in marketing and consumer behavior, but also for psychologists, educators, sociologists, and policymakers.

**Author Contributions:** Conceptualization, K.-T.L.; Methodology, K.-T.L.; Software, K.-T.L.; Validation, K.-T.L.; Formal Analysis, K.-T.L.; Investigation, K.-T.L. and H.F.; Resources, K.-T.L. and H.F.; Data Curation, K.-T.L. and H.F.; Writing—Original Draft Preparation, K.-T.L.; Writing—Review & Editing, K.-T.L. and H.F.; Visualization, K.-T.L.; Project Administration, K.-T.L. and H.F.; Funding Acquisition, K.-T.L. and H.F. All authors have read and agreed to the published version of the manuscript.

**Funding:** This research was funded by JSPS KAKENHI grant number JP20K13623. The APC was funded by the Chuo University Personal Research Grant.

**Institutional Review Board Statement:** Ethical review and approval were waived for this study because of institutional legality and key informant approval.

**Informed Consent Statement:** Respondents provided informed consent for using the data by answering the questionnaire. The online survey was conducted as follows: participation was voluntary, anonymity was guaranteed, and respondents could not only refuse to participate, but also quit answering questions at any time.

**Data Availability Statement:** The data presented in this study are available upon reasonable request from the corresponding authors.

**Conflicts of Interest:** The authors declare no conflict of interest.

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
