# Peer review of "Exploring Subjective Happiness, Life Satisfaction, and Sustainable Luxury Consumption in China and Japan Amidst the COVID-19 Pandemic"

_admsci, doi:10.3390/admsci13070169_

Round 1

Reviewer 1 Report

The authors conducted a cross-national investigation to examine the relationship between subjective well-being (SWB) and sustainable luxury consumption, considering subjective happiness (SH), life satisfaction (LS), and consumer novelty seeking (CNS). The authors applied a conceptual model and analyzed the data collected from 800 respondents in China and Japan during the pandemic. The authors claim that partial least squares structural equation modeling results derive novel findings: SWB brings about pro-environmental behavioral intentions even when consuming luxury fashion products. Although SH and LS represent SWB, their effects on luxury consumption vary across China and Japan. In particular, the impact of the SH was more pronounced in China than in Japan. For the authors, CNS mediates the effects of SH and LS on sustainable luxury consumption.

The article is well-written and presents a structure that meets the proposed objective, bringing insights into the marketing area, specifically about consumer behavior.

In the current phase of the article, given the corrections and adjustments suggested in the JHC, the article has improved considerably and has the potential to be published.

Reviewer 2 Report

Dear authors,

Your paper is very interesting and promising. However, I would like to suggest some changes and improvements.

Title: The title is perhaps the only part of your paper that most readers will read. Please consider the improved version of the journal article title:

"Exploring Subjective Happiness, Life Satisfaction, and Sustainable Luxury Consumption in China and Japan Amidst the COVID-19 Pandemic: Unraveling the Impact of Consumer Novelty Seeking"

1)      Conciseness: The original title is quite long and includes multiple concepts and variables.

2)      Engaging language: By incorporating words like "exploring" and "unraveling," we create a sense of curiosity and intrigue, enticing readers to delve into the study.

3)      Clarity: The revised title explicitly mentions that the research is conducted "amidst the COVID-19 pandemic." This inclusion helps establish the context and relevance of the study.

4)      Emphasis on sustainable luxury consumption: The original title does not explicitly highlight the focus on sustainable luxury consumption.

5)      Mediating role of consumer novelty seeking: The revised title emphasizes the mediating role of consumer novelty seeking.

Abstract:

Some readers would appreciate additional information in your abstract. I would like to suggest some changes:

The original introduction should be expanded upon to provide a clearer overview of the research.

One sentence about analyzing data from 800 respondents in China and Japan during the pandemic should be included to highlight the scope of the study and the specific context in which the research was conducted.

The purpose of the study should be clarified, stating that the goal is to shed light on these relationships and uncover novel findings. This helps readers understand the objectives and the potential contributions of the research.

The specific finding that subjective well-being (SWB) positively influences pro-environmental behavioral intentions, even in the context of luxury fashion consumption, could be explicitly mentioned. This highlights a key result of the study and reinforces the relevance of the research.

The variation in the impact of subjective happiness (SH) and life satisfaction (LS) on luxury consumption between China and Japan could be emphasized. By specifying that the influence of SH was more pronounced in China than in Japan, the authors draw attention to a unique finding that may be of interest to readers.

The role of consumer novelty seeking (CNS) as a mediator in the effects of SH and LS on sustainable luxury consumption could be highlighted. This finding adds another layer of understanding to the relationships being explored in the study.

Introduction:

About the reviewed literature, many of the references cited in the literature review are quite outdated. It is recommended that no more than 50% of the references be older than five years. Additionally, it is suggested to consider publications from 2023 that focus on this topic or target group, such as IJERPH, Sustainability, Behavioral Sciences, and Education Sciences, among others.

Objective:

The objective of the study is not formulated. Furthermore, the advantages and benefits of conducting this research are not sufficiently explained.

Regarding the text between lines 102-108, which is crucial because it explains the research questions, please clarify the objective of the study by adding a statement indicating that the study aims to address the research gap.

Method:

The section regarding the methodology could be improved. It should include more information about the data collection procedure, response rate, the ethical committee that approved the project, and the treatment of missing data, outliers, etc. I cannot exactly understand HOW the surveys have been distributed. And, if they were a consequence of a luxury acquisition made by the respondent, or were independent.

Statistical analysis:

The procedures of SEM analyses are highly suitable. The results are presented clearly and rigorously.

Discussion:

Finally, it would be beneficial to expand the section on limitations, future avenues of research, and the key contributions of the study. It is important to acknowledge that the sample size is obtained through the Internet. Can you ascertain that all participants are true as they inform? Is there any method to ensure the accuracy of the information provided by participants regarding their age, gender, etc.? Your paper holds significant implications for educators, psychologists, society, and policymakers; however, further elaboration on this topic is needed.

General comments:

Throughout the entire text, there are numerous abbreviations and acronyms, which hinder readers' comprehension. Please, minimize the use of such abbreviations.
